# One-Step Synthesis of Ag_2_O/Fe_3_O_4_ Magnetic Photocatalyst for Efficient Organic Pollutant Removal via Wide-Spectral-Response Photocatalysis–Fenton Coupling

**DOI:** 10.3390/molecules28104155

**Published:** 2023-05-17

**Authors:** Chuanfu Shan, Ziqian Su, Ziyi Liu, Ruizheng Xu, Jianfeng Wen, Guanghui Hu, Tao Tang, Zhijie Fang, Li Jiang, Ming Li

**Affiliations:** 1College of Science & Key Laboratory of Low-Dimensional Structural Physics and Application, Education Department of Guangxi Zhuang Autonomous Region, Guilin University of Technology, Guilin 541004, China; 2School of Electronics Engineering, Guangxi University of Science and Technology, Liuzhou 545006, China

**Keywords:** photocatalysis, Ag_2_O/Fe_3_O_4_, heterosuperstructure, Fenton reaction, magnetic separation

## Abstract

Photocatalysis holds great promise for addressing water pollution caused by organic dyes, and the development of Ag_2_O/Fe_3_O_4_ aims to overcome the challenges of slow degradation efficiency and difficult recovery of photocatalysts. In this study, we present a novel, environmentally friendly Ag_2_O/Fe_3_O_4_ magnetic nanocomposite synthesized via a simple coprecipitation method, which not only constructs a type II heterojunction but also successfully couples photocatalysis and Fenton reaction, enhancing the broad-spectrum response and efficiency. The Ag_2_O/Fe_3_O_4_ (10%) nanocomposite demonstrates exceptional degradation performance toward organic dyes, achieving 99.5% degradation of 10 mg/L methyl orange (MO) within 15 min under visible light irradiation and proving its wide applicability by efficiently degrading various dyes while maintaining high stability over multiple testing cycles. Magnetic testing further highlighted the ease of Ag_2_O/Fe_3_O_4_ (10%) recovery using magnetic force. This innovative approach offers a promising strategy for constructing high-performance photocatalytic systems for addressing water pollution caused by organic dyes.

## 1. Introduction

With the acceleration of global industrialization, water pollution caused by organic dyes has become an increasingly urgent issue of public concern. Photocatalytic technology has been widely researched owing to its advantages, such as high efficiency and the absence of secondary pollution [1,2,3,4,5,6,7,8] Among them, silver oxide (Ag_2_O) nanoparticles are extensively employed to degrade water pollutants due to their simple preparation, stable properties, and environmental friendliness [9,10,11,12]. However, the narrow bandgap and low photogenerated carrier-separation efficiency of Ag_2_O limit its application in water treatment [13,14,15,16]. Furthermore, using Ag_2_O as a powder photocatalyst makes it challenging to recycle. Therefore, it is practically significant to develop a magnetic material to couple with Ag_2_O through heterojunctions to enhance the photogenerated carrier separation efficiency and fabricate an efficient and magnetically recoverable Ag_2_O-based photocatalyst.

Fe_3_O_4_ nanoparticles possess unique magnetic properties and nanoscale characteristics, which make them highly versatile for various potential applications, including drug delivery, MRI contrast agents, biomolecule separation, biosensing, and catalysis, due to their magnetism and biocompatibility [17,18,19,20,21,22]. Additionally, Fe_3_O_4_ nanoparticles demonstrate strong light-absorption ability, absorbing most of the light in the UV–visible range [23]. Furthermore, the Fe^2+^ ions in Fe_3_O_4_ can react with hydrogen peroxide (H_2_O_2_) through the Fenton reaction to generate a large number of free radical groups, which can oxidize many known organic compounds, such as carboxylic acids, alcohols, and esters, into inorganic forms, exhibiting a significant oxidation ability to remove refractory organic pollutants [24,25,26,27]. It is known that the Fenton reaction can produce a large number of oxygen-related species through the following reactions:(1)$$2{Fe}^{2+}+H_{2}O_{2}+O_{2}\to 2{Fe}^{3+}+{\cdot O}_{2}^{-}+(OH{)}^{-}+\cdot OH$$
(2)$$H_{2}O_{2}+2Fe^{3+}\to 2{Fe}^{2+}+O_{2}+2H^{+}$$

Additionally, Fe_3_O_4_ is a magnetic material that can be recycled and reused by an external magnetic field, thereby reducing the cost of recovery treatment [28,29,30]. In contrast, the use of Fe_3_O_4_ in the Fenton reaction to oxidize organic pollutants in water has significant limitations, including the need to consume externally provided H_2_O_2_ in the reaction and the requirement of an acidic environment to generate free radicals through the Fenton reaction. However, when Fe_3_O_4_ is coupled with a photocatalyst, the H_2_O_2_ generated at the interface of the photocatalyst can be used for the Fenton reaction with Fe_3_O_4_. Another advantage of selecting Fe_3_O_4_ is that it contains both Fe^2+^ and Fe^3+^ ions, facilitating the continuous progress of the Fenton reaction. Consequently, numerous researchers have employed Fe_3_O_4_ as a cocatalyst in the photocatalyst system [31,32,33,34].

In this article, the Ag_2_O/Fe_3_O_4_ binary magnetic nanoparticles were synthesized using a simple chemical coprecipitation method with FeCl_2_·4H_2_O, FeCl_3_·6H_2_O, and AgNO_3_ as raw materials, and they were applied to degrade the organic dyes in water. The results showed that the introduction of Fe_3_O_4_ to load Ag_2_O could generate a type II heterojunction at the contact interface, facilitating the fast transfer of photogenerated carriers. At the same time, the photocatalysis–Fenton combined reaction was also constructed to improve the utilization efficiency of photogenerated carriers, further enhancing the degradation efficiency of the photocatalyst. The Ag_2_O/Fe_3_O_4_ nanoparticles exhibited very high efficiency in degrading dyes, such as methyl orange (MO), under visible light irradiation.

## 2. Results and Discussion

### 2.1. TEM Analysis

Transmission electron microscopy (TEM) was used to characterize the microstructure of the samples. Figure 1 shows the results. Firstly, TEM analysis was performed on Fe_3_O_4_ nanoparticles, and the average particle size was found to be 15 ± 5 nm with a typical spherical morphology, as shown in Figure 1a. Due to the large surface area, the Fe_3_O_4_ nanoparticles exhibited obvious aggregation in the image. Figure 1b shows the TEM image of the Ag_2_O nanoparticles, which had a particle size distribution ranging from 30 to 80 nm and a polyhedral morphology that differed greatly from Fe_3_O_4_. The nanoparticles of the two materials could be easily distinguished. Figure 1c shows the TEM image of binary Ag_2_O/Fe_3_O_4_ (10%) nanoparticles, which demonstrates that Fe_3_O_4_ nanoparticles with smaller size and spherical morphology could encapsulate Ag_2_O nanoparticles with larger size and polyhedral morphology, indicating good compatibility between Fe_3_O_4_ and Ag_2_O. A high-resolution TEM (HRTEM) analysis was performed on a circular region indicated in Figure 1c to confirm the successful formation of Ag_2_O/Fe_3_O_4_ (10%). Figure 1d shows the results. The clear interface between Fe_3_O_4_ nanoparticles and Ag_2_O nanoparticles was observed with lattice spacing of 0.25 nm (corresponding to the (311) plane of/Fe_3_O_4_) and 0.29 nm (corresponding to the (220) plane of Ag_2_O), respectively, further demonstrating the successful formation of Ag_2_O/Fe_3_O_4_ (10%).

### 2.2. SEM and EDS Analysis

Ag_2_O/Fe_3_O_4_ (10%) was analyzed through scanning electron microscopy to better verify the successful coupling of Ag_2_O and Fe_3_O_4_ to form Ag_2_O/Fe_3_O_4_ binary nanoparticles, as shown in Figure 2. A relatively large size was selected for characterization to better analyze the overall morphology and surface element distribution of the binary nanoparticles. As shown in Figure 2a, it can be observed that the smaller Fe_3_O_4_ nanoparticles with approximately spherical shape were well loaded onto the surface of larger-sized Ag_2_O nanoparticles with polyhedral shape, forming a compact structure of binary nanoparticles with good structural stability, which is consistent with the conclusion obtained from the TEM image analysis. Additionally, the loading of Fe_3_O_4_ increased the number of reaction sites. Note that almost no Fe_3_O_4_ spherical nanoparticles were present in unoccupied areas on the surface of Ag_2_O, indicating that Ag_2_O has a good ability to capture Fe_3_O_4_. EDS analysis was performed in this area to analyze the surface element distribution of Ag_2_O/Fe_3_O_4_ (10%) binary nanoparticles. Figure 2b–e shows the results, where all Fe elements of Fe_3_O_4_ are uniformly distributed on the surface of Ag_2_O, indicating that Fe_3_O_4_ was successfully loaded onto the surface of Ag_2_O to form Ag_2_O/Fe_3_O_4_ binary nanoparticles. EDS data statistics were conducted to further demonstrate the contents of Fe_3_O_4_ and Ag_2_O. Table 1 presents the results. It can be seen that ratio of the number of Ag atoms and Fe atoms is approximately 6:1, indicating that the mass ratio of Ag_2_O to Fe_3_O_4_ is approximately 9:1, and Fe_3_O_4_ accounts for 10% of the total mass.

### 2.3. XRD Analysis

X-ray diffraction (XRD) was used to characterize the pure Ag_2_O and Fe_3_O_4_ nanoparticles as well as Ag_2_O/Fe_3_O_4_ (10%) nanocomposites to investigate their crystal structure. Figure 3 shows the results. The diffraction peaks of Ag_2_O nanoparticles at X-ray diffraction angles (2θ) of 26.9°, 33.0°, 38.3°, 55.1°, 65.7°, and 68.7° were indexed to the (110), (111), (200), (220), (311), and (222) crystal planes of Ag_2_O, respectively, which were consistent with the JCPDS card (PDF#75-1532) for Ag_2_O. The diffraction peaks of Fe_3_O_4_ nanoparticles at X-ray diffraction angle (2θ) of 28.26°, 34.53°, 44.01° and 61.88° were indexed to the (220), (311), (400), and (440) crystal planes of Fe_3_O_4_, respectively, which were consistent with the JCPDS card (PDF#19-0629) for Fe_3_O_4_. The XRD pattern of Ag_2_O/Fe_3_O_4_ (10%) nanocomposites showed the same diffraction peaks at 26.9°, 33.0°, 38.3°, 55.1°, 65.7°, and 68.7° for Ag_2_O and at 28.26°, 34.53°, 44.01°, and 61.88° for Fe_3_O_4_, indicating a good coupling of Ag_2_O and Fe_3_O_4_ and showing no change in their crystal structure. Additionally, no other impurity phases were observed, indicating that Ag_2_O/Fe_3_O_4_ (10%) is a two-phase composite.

### 2.4. XPS Elemental Analysis

X-ray photoelectron spectroscopy (XPS) was used to investigate the chemical composition of the Ag_2_O/Fe_3_O_4_ (10%) sample. Figure 4a shows the XPS spectrum of the sample, exhibiting distinct peaks at around 285.2 eV (C 1s), 368.8 eV (Ag 3d), 530.08 eV (O 1s), and 711.08 eV (Fe 2p), which indicate the presence of four elements, namely C, Ag, O, and Fe. The presence of C is attributed to the fixation of CO_2_ from air during the preparation of the binary composite material. XPS fine-spectrum measurement was performed to investigate the elemental state in detail. Figure 4b shows the Ag 3d fine spectrum, exhibiting binding energies of 368.2 eV and 374.0 eV for Ag 3d5/2 and Ag 3d3/2, respectively. These binding energies correspond to the orbit peaks of Ag^+^ in Ag_2_O, confirming the existence of Ag_2_O in the compound. As depicted in Figure 4c, the Fe 2p XPS spectrum reveals two spin–orbit doublets. The first doublet, attributed to Fe^2+^, is observed at 710.58 eV (Fe 2p3/2) and 723.78 eV (Fe 2p1/2), while the second doublet, assigned to Fe^3+^, is observed at 712.18 eV (Fe 2p3/2) and 726.12 eV (Fe 2p1/2). This mixed phase confirms the formation of Fe_3_O_4_. Figure 4d shows the O 1s fine spectrum, in which the peak at 532.11 eV is attributed to external −OH groups or adsorbed water molecules on the surface, the peak at 531.11 eV corresponds to the lattice oxygen atoms in Ag_2_O, and the peak at 529.39 eV is attributed to the Fe-O bond [35,36]. Therefore, XPS analysis confirms the presence of Ag_2_O and Fe_3_O_4_ in the Ag_2_O/Fe_3_O_4_ (10%) binary nanocomposite material and their successful composition.

### 2.5. UV–Vis and PL Analysis

UV*–*vis and PL tests were conducted to determine the optical properties of the synthesized nanomaterials. UV*–*vis testing was used to measure the absorbance of the synthesized nanomaterials. Figure 5a shows the results. Ag_2_O exhibits strong absorption in the ultraviolet and near-ultraviolet regions, with a peak at a wavelength of 500 nm [13,14,15]. Fe_3_O_4_ exhibits a strong optical response across the whole examined spectral range, indicating that the strong visible light-absorption capability of Ag_2_O/Fe_3_O_4_ binary composite catalysts is undoubtedly due to the optical properties of Fe_3_O_4_ [23,31,32,33]. Furthermore, compared with Ag_2_O, a gradual redshift was observed at the absorption edge of Ag_2_O/Fe_3_O_4_ binary composite catalysts, and a significant increase in absorption was observed in the near-infrared region of 600–800 nm, indicating a strong interaction between Ag_2_O and Fe_3_O_4_ in the binary composite catalyst. It is worth noting that as the loading amount of Fe_3_O_4_ increases, the light absorption ability of Ag_2_O/Fe_3_O_4_ binary photocatalysts in the UV*–*visible spectral range also increases. Ag_2_O/Fe_3_O_4_ (15%) exhibits the best light-absorption ability, followed by Ag_2_O/Fe_3_O_4_ (10%) and Ag_2_O/Fe_3_O_4_ (5%). The Kubelka–Munk equation was used to calculate the bandgap energy of the semiconductor:(3)$${\left(ahv\right)}^{\frac{2}{n}}=A\left(hv-E_{g}\right)$$
where *α* represents the absorption coefficient of the semiconductor, *h* is a constant and stands for the Planck constant, *v* represents the frequency of light, *A* is a constant and represents a constant term, and *n* is closely related to the semiconductor transition process. The indirect semiconductors Ag_2_O and Fe_3_O_4_ both have n values of 4. The Kubelka-Munk function was used to derive the absorption spectra of all the synthesized catalysts, which were then used to generate Tauc plots. As shown in Figure 5b, the results of Tauc plots show that the optical bandgaps of Ag_2_O and Fe_3_O_4_ are 2.0 eV and 1.2 eV, respectively, while the bandgaps of Ag_2_O/Fe_3_O_4_ (5%), Ag_2_O/Fe_3_O_4_ (10%), and Ag_2_O/Fe_3_O_4_ (15%) are 1.6 eV, 1.5 eV, and 1.4 eV, respectively. These results are in good agreement with the increasing trend in the redshift observed at the absorption edge with the increase in the Fe_3_O_4_ loading amount shown in Figure 5a.

When Ag_2_O and Fe_3_O_4_ are exposed to light, valence band electrons absorb photon energy and transition to the conduction band, forming photogenerated electron–hole pairs. PL emission occurs when conduction band electrons recombine with valence band holes. Therefore, PL intensity is proportional to the separation of photogenerated charge carriers; lower PL intensity reflects a reduction in recombination probability. As shown in Figure 5c, when the samples of Ag_2_O, Ag_2_O/Fe_3_O_4_ (5%), Ag_2_O/Fe_3_O_4_ (10%), and Ag_2_O/Fe_3_O_4_ (15%) were subjected to PL testing under 260nm excitation light, their emission peak positions were all at 400 nm. Although the emission peak intensity of Ag_2_O was higher, it decreased with the loading of Fe_3_O_4_. Compared with Ag_2_O, the emission peak intensity of Ag_2_O/Fe_3_O_4_ (5%) decreased to 90%, while that of Ag_2_O/Fe_3_O_4_ (10%) decreased significantly to 60%. However, as the loading of Fe_3_O_4_ continued to increase, the emission peak intensity of Ag_2_O/Fe_3_O_4_ (15%) was higher than that of the original Ag_2_O. Therefore, it can be concluded that photogenerated electron–hole pairs are generated when light is irradiated onto the surface of Ag_2_O. When Fe_3_O_4_ with a low loading is coupled to the surface of Ag_2_O, they enhance the separation efficiency of photogenerated electron–hole pairs generated by Ag_2_O. However, when the loading of Fe_3_O_4_ exceeds 15% of the total mass, an excess Fe_3_O_4_ forms a thick covering layer on the surface of Ag_2_O. Under light irradiation, Fe_3_O_4_ absorbs photons, causing the electrons on the valence band to be excited to the conduction band, generating photogenerated electron–hole pairs. Due to the low optical bandgap of Fe_3_O_4_, which is only 1.6 eV, photogenerated electrons and holes are prone to recombine, producing strong light emission.

### 2.6. Electrochemical Characterization Analysis

Mott–Schottky analysis, photocurrent response analysis, and EIS were performed to determine the electrochemical properties of the prepared samples. The Mott–Schottky plot is the most commonly used method to distinguish between n-type and p-type semiconductors [37]. A positive slope and a negative slope indicate an n-type and a p-type semiconductor, respectively. Additionally, the Mott–Schottky plot can be extrapolated to estimate the flat-band potential (Efb) of the semiconductor, which can be used to estimate the position of the Fermi level [38]. Assuming that the Fermi level is very close to the band edge, the extrapolated flat-band potential (Efb) can be utilized as the position of the edge of either the n-type semiconductor (E_CB_) or the p-type semiconductor (E_VB_). Figure 6a,b shows the Mott–Schottky plots of Ag_2_O and Fe_3_O_4_ with Ag/AgCl as the reference electrode. It can be seen that both Ag_2_O and Fe_3_O_4_ have positive slopes, indicating that they are p-type semiconductors. By extrapolation, the Mott–Schottky plots of Ag_2_O and Fe_3_O_4_ intersect the *x*-axis at 1.84 eV and 1.95 eV, respectively. Considering the difference between the reference electrode (Ag/AgCl) and the standard value of 0.19 eV (relative to the normal hydrogen electrode), the Evb values of Ag_2_O and Fe_3_O_4_ are estimated to be 2.03 eV and 2.14 eV, respectively. Furthermore, based on the previously obtained data, the optical bandgaps of Ag_2_O and Fe_3_O_4_ are 2.0 eV and 1.2 eV, respectively. Therefore, the E_VB_ of Ag_2_O and Fe_3_O_4_ can be calculated using the following equation:(4)$$E_{VB}=E_{g}+E_{CB}$$
where E_g_ represents the optical bandgap energy. By substituting the values of E_g_ as 2.0 eV and E_VB_ as 2.03 eV for Ag_2_O in the formula, the value of E_CB_ is calculated as −0.03 eV. Similarly, by substituting the values of E_g_ as 1.2 eV and E_VB_ as 2.14 eV for Fe_3_O_4_ in the formula, the value of E_CB_ is calculated as 0.94 eV. The photogenerated current response analysis can be used to verify the efficiency of the photogenerated carriers in the samples. The research results show that when a small amount of Fe_3_O_4_ is loaded on the surface of Ag_2_O, the photocurrent intensity of the sample is significantly improved, and the photocurrent intensity of Ag_2_O/Fe_3_O_4_ (10%) is the highest, as shown in Figure 6c. However, when the loading amount of Fe_3_O_4_ reaches 15%, the photocurrent intensity of the formed Ag_2_O/Fe_3_O_4_ (15%) is lower than that of Ag_2_O. This indicates that too much Fe_3_O_4_ loading will reduce the utilization efficiency of photogenerated carriers. In addition, EIS measurements were also conducted to study the charge-transfer resistance and transfer efficiency of photogenerated carriers. As shown in Figure 6d, it can be observed that the Nyquist semicircle diameters of the Ag_2_O/Fe_3_O_4_ (5%) and Ag_2_O/Fe_3_O_4_ (10%) nanocomposites are smaller than those of Ag_2_O and Ag_2_O/Fe_3_O_4_ (15%). The Nyquist semicircle diameter of Ag_2_O/Fe_3_O_4_ (10%) is the lowest, indicating that its resistance is lower than that of Ag_2_O and the other samples. Therefore, loading a small amount of Fe_3_O_4_ can improve the transfer efficiency of photogenerated carriers in Ag_2_O, which is a favorable condition for enhancing the photocatalytic activity. However, when the loading amount of Fe_3_O_4_ reaches 15%, the Nyquist semicircle diameter of Ag_2_O/Fe_3_O_4_ (15%) is larger than that of Ag_2_O, indicating that too much Fe_3_O_4_ loading will reduce the available surface area of oxidized silver, leading to an increase in the resistance encountered by electrons and holes during transmission and a decrease in the transfer efficiency of photogenerated carriers.

### 2.7. Photocatalytic Performance Analysis

Fe_3_O_4_, Ag_2_O, Ag_2_O/Fe_3_O_4_ (5%), Ag_2_O/Fe_3_O_4_ (10%), and Ag_2_O/Fe_3_O_4_ (15%) were placed under a xenon lamp light source (λ > 420nm) to simulate visible light in sunlight and to photocatalyze a 10 mol/L MO solution to better demonstrate the visible light photocatalytic performance of different samples. Figure 7a shows the results. When pure Fe_3_O_4_ was placed in the MO solution and irradiated with visible light, no MO degradation was observed, indicating that Fe_3_O_4_ alone does not have the ability to degrade the MO solution under visible light. When pure Ag_2_O was placed in the MO solution and irradiated with visible light, MO was significantly degraded. Approximately 80% of the MO solution was degraded in 15 min of light irradiation, and 99.1% of the MO solution was degraded in 30 min of light irradiation, indicating that Ag_2_O can absorb photon energy and produce photocatalytic reactions under visible light, which is a considerable catalytic rate for the MO solution.

The photocatalytic rate of Ag_2_O changed significantly after Fe_3_O_4_ was loaded onto its surface. When the Fe_3_O_4_ loading amount was 5wt% of the overall weight, the Ag_2_O/Fe_3_O_4_ (5%) binary catalyst was formed, degrading 99.1% of the MO solution after 15 min of visible light irradiation. When the Fe_3_O_4_ loading amount was 10wt% of the overall weight, the Ag_2_O/Fe_3_O_4_ (10%) binary catalyst was formed, degrading 99.5% of the MO solution after 15 min of visible light irradiation. It is worth noting that when the Fe_3_O_4_ loading amount continued to increase to 15wt% of the overall weight, the Ag_2_O/Fe_3_O_4_ (15%) binary catalyst did not increase but decreases the degradation rate of the MO solution. Moreover, it only degraded 75.1% and 85.2% of the MO solution after 15 min and 30 min of visible light irradiation, respectively. This indicates that during the Fe_3_O_4_ loading, a coverage layer forms on the surface of Ag_2_O, and Fe_3_O_4_ absorb photons and produce electron–hole pairs under light irradiation, which are then be transferred from type-II heterojunction to the electrode of Ag_2_O to participate in the reaction. However, when there is too much Fe_3_O_4_ loading, the thicker coverage layer it forms reduces the available surface area of Ag_2_O, blocks the entry of photons, and shields the surface of Ag_2_O from light, thereby reducing the generation of photoinduced carriers. In addition, in photocatalytic reactions, electrons and holes are transmitted through the surface conductor, thereby participating in redox reactions. The reduction in the available surface area of oxidized silver increases the resistance encountered by electrons and holes during transmission, resulting in a slower charge transfer rate and reduced reaction efficiency under visible light irradiation. Figure 7b shows the UV–vis absorption spectra during the photocatalytic degradation of the MO solution using Ag_2_O/Fe_3_O_4_ (10%). It can be observed that the absorption peak at 464 nm of MO decreases significantly with the irradiation time, and the peak intensity almost reaches zero after 15 min of irradiation. No new absorption peaks were generated, indicating that MO was completely degraded into inorganic substances without the formation of other organic compounds. Figure 7c shows the degradation of MO by different photocatalysts using a pseudo-first-order kinetics model. It can be seen that the degradation rate of Ag_2_O/Fe_3_O_4_ (10%) is the fastest, reaching 0.183 min^−1^, which is 2.3 times higher than that of pure Ag_2_O (0.078 min^−1^), 3 times higher than that of Ag_2_O/Fe_3_O_4_ (15%) (0.061 min^−1^), and 1.17 times higher than that of Ag_2_O/Fe_3_O_4_ (5%) (0.156 min^−1^). Four photocatalytic cycling tests were conducted to verify the structural stability of the Ag_2_O/Fe_3_O_4_ (10%) sample. Figure 7d shows the results. After four cycles, the catalytic rate of Ag_2_O/Fe_3_O_4_ (10%) slightly decreased but still exhibited a fast catalytic rate, indicating good structural stability.

In general, the reactive species in photocatalytic processes are often considered to be holes (h^+^), hydroxyl radicals (·OH), and superoxide ion radicals (·O_2_^−^). Therefore, EDTA-2Na, isopropyl alcohol (IPA), and benzene quinone (BQ) were selected as the capture agents to study the capture of these reactive species, as shown in Figure 7e. Through the photodegradation experiment of MO using the Ag_2_O/Fe_3_O_4_ (10%) photocatalyst under visible light, in which the original photocatalytic degradation was 18.3 × 10^−2^ min^−1^, it was observed that the degree of inhibition of the photocatalytic degradation rate decreased in the following order: BQ (1.85 × 10^−2^ min^−1^), IPA (2.76 × 10^−2^ min^−1^), and EDTA-2Na (11.2 × 10^−2^ min^−1^). This reveals that ·O_2_^−^ and ·OH have a significant impact on the degradation of MO in the Ag_2_O/Fe_3_O_4_ photocatalytic reaction, while h^+^ has a relatively small degree of participation.

The catalytic rates of phenol, rhodamine B, methyl blue, and basic fuchsin were tested under visible light irradiation to verify the applicability of the Ag_2_O/Fe_3_O_4_ (10%) photocatalyst for the degradation of organic pollutants in water. As shown in Figure 7f, the degradation rates of basic fuchsin, rhodamine B, and methyl blue were 10.72 × 10^−2^ min^−1^, 9.37 × 10^−2^ min^−1^, and 6.1 × 10^−2^ min^−1^, respectively. This demonstrates that the Ag_2_O/Fe_3_O_4_ (10%) photocatalyst has a good applicability and a good catalytic effect on various types of organic pollutants in water.

### 2.8. Magnetic Properties Analysis

Vibrating sample magnetometer (VSM) measurements were performed on Ag_2_O/Fe_3_O_4_ (5%), Ag_2_O/Fe_3_O_4_ (10%), and Ag_2_O/Fe_3_O_4_ (15%) to determine the magnetic properties of the samples. Figure 8a–c show the results. As shown in Figure 8a, the magnetic properties of the Ag_2_O/Fe_3_O_4_ binary nanocomposites gradually increase with the increase in Fe_3_O_4_ surface loading content. Ag_2_O/Fe_3_O_4_ (15%) exhibited the strongest magnetism, with a maximum saturation magnetization of 1.01 emu/g and a hysteresis loop showing a clear bent shape without a saturation region. In contrast, Ag_2_O/Fe_3_O_4_ (10%) and Ag_2_O/Fe_3_O_4_ (5%) exhibited the maximum saturation magnetization of 0.31 emu/g and 0.15 emu/g, respectively. Figure 8b,c show the hysteresis loop characteristics of Ag_2_O/Fe_3_O_4_ (5%) and Ag_2_O/Fe_3_O_4_ (10%), respectively. It can be observed that although the size of the magnetic moment increases with the external magnetic field, its maximum value is much smaller than the saturation magnetization of ferromagnetic materials. Therefore, the hysteresis loop shows a curve similar to paramagnetism. Since the magnetic moment is very small, the hysteresis loop of Ag_2_O/Fe_3_O_4_ nanocomposites is smoother and more symmetrical than that of paramagnetic materials. Therefore, due to the introduction of Fe_3_O_4_, it was confirmed that Ag_2_O/Fe_3_O_4_ (5%), Ag_2_O/Fe_3_O_4_ (10%), and Ag_2_O/Fe_3_O_4_ (15%) all have superparamagnetic properties [39].

A neodymium magnet adsorption experiment was performed to verify whether Ag_2_O/Fe_3_O_4_ (10%) can be magnetically recovered. Figure 8d shows the results. Specifically, a neodymium magnet was placed next to the Ag_2_O/Fe_3_O_4_ (10%) suspension and was allowed to stand still for 20 min. It was observed that the neodymium magnet clearly adsorbed the gray-brown catalyst powder. Therefore, it was confirmed that magnetic adsorption can recover Ag_2_O/Fe_3_O_4_ (10%).

### 2.9. Photocatalytic Reaction Mechanism Analysis

First, under visible light irradiation, Ag_2_O and Fe_3_O_4_ on the surface of Ag_2_O/Fe_3_O_4_ are excited from the valence band to the conduction band, generating photogenerated electrons (e^−^) and leaving behind holes (h^+^). As the E_CB_ of Ag_2_O is −0.67 eV, which is more negative than that of Fe_3_O_4_, i.e., 0.54 eV, and the E_VB_ of Ag_2_O is 2.03 eV, which is more negative than that of Fe_3_O_4_, i.e., 2.14 eV, a type-II heterojunction is formed due to the band offset when the two are coupled. The h^+^ on the Fe_3_O_4_ valence band transfers to the Ag_2_O valence band, and the e^-^ on the Ag_2_O conduction band transfers to the Fe_3_O_4_ conduction band, thus improving the separation efficiency of the photogenerated electrons and holes. These electrons and holes then participate in other reactions. The e^-^ on the Fe_3_O_4_ conduction band reacts with the dissolved oxygen and water in the liquid to form H_2_O_2_ and OH^−^. H_2_O_2_ can then further participate in the Fenton reaction, while the h^+^ on the Ag_2_O valence band reacts with H_2_O to generate ·OH free radicals and H^+^.

Second, the Fenton reaction occurs on Fe_3_O_4_. Fe^2+^ in Fe_3_O_4_, H_2_O_2_ generated in the photocatalytic reaction, and dissolved O_2_ in water react to generate Fe^3+^, ·O_2_^−^, and ·OH, respectively. Next, H_2_O_2_ can reduce Fe^3+^ to replenish the consumed Fe^2+^ and O_2_ and generate H^+^, so the reaction can be cycled. The large amounts of ·OH, and ·O_2_^−^ generated by the combined photocatalytic and Fenton reactions can further participate in the oxidation and degradation of organic compounds, decomposing them into smaller harmless compounds, as shown in Figure 9. The specific reaction process is as follows:(5)$${Ag}_{2}O/{Fe}_{3}O_{4}+hv\to e^{-}+h^{+}$$
(6)$$h^{+}+H_{2}O\to \cdot OH+H^{+}$$
(7)$$2e^{-}+O_{2}+2H_{2}O\to H_{2}O_{2}+2OH^{-}$$
(8)$$\cdot O_{2}+MO\to Degraded\;products$$
(9)$$H_{2}O_{2}+2{Fe}^{3+}\to {2Fe}^{2+}+O_{2}+2H^{+}$$
(10)$$\cdot O_{2}+MO\to Degraded\;products$$
(11)·OH+MO→Degraded products

## 3. Materials and Methods

### 3.1. Material

Silver nitrate (AgNO_3_, 99%) was purchased from National Pharmaceutical Group Co., Ltd. (Shanghai, China). Iron(II) chloride tetrahydrate (FeCl_2_·4H_2_O, ≥92.0%), Iron(III) chloride hexahydrate (FeCl_3_·6H_2_O, ≥98.1%), sodium hydroxide (NaOH, ≥96.0%), and methyl orange (MO) were purchased from Xilong Science Co., Ltd. (Guangdong, China). All the raw materials were of analytical grade and used without any additional purification. Deionized water was used for all the experiments.

### 3.2. Preparation of Fe_3_O_4_

Approximately 0.198 g of FeCl_3_·6H_2_O and 0.072 g of FeCl_2_·4H_2_O were dissolved in 200 mL of deionized water by ultrasonication for 30 min. Approximately 20 mL of NaOH solution (1 M) was then added. The mixture was sonicated for 1 h, centrifuged, washed three times with deionized water, and then freeze-dried to obtain Fe_3_O_4_.

### 3.3. Preparation of Ag_2_O

Approximately 0.5 g of AgNO_3_ was dissolved in 200 mL of deionized water, and 20 mL of NaOH solution (1 M) was then added. The mixture was sonicated for 1 h, centrifuged, washed three times with deionized water, and then freeze-dried to obtain Ag_2_O.

### 3.4. Preparation of Ag_2_O/Fe_3_O_4_

The Ag_2_O/Fe_3_O_4_ catalyst was prepared using a one-step coprecipitation method. First, 0.5 g of AgNO_3_ was dissolved ultrasonically in 200 mL of deionized water, and 0.041, 0.088, and 0.119 g of FeCl_3_·6H_2_O were dissolved with 0.015, 0.032, and 0.044 g of FeCl_2_·4H_2_O, respectively, in 200 mL of deionized water as different precursors of Fe_3_O_4_. The precursor of the Ag_2_O solution was then added into the Fe_3_O_4_ precursor solutions and treated ultrasonically for 1 h. Next, a 1 M NaOH solution was continuously dripped into the quickly stirred precursor solution until no further color change was observed. Finally, the product was washed three times by centrifugation, freeze-dried, and obtained as Ag_2_O/Fe_3_O_4_ (5%, 10%, 15%) binary photocatalysts.

### 3.5. Characterization

The samples were subjected to various analytical techniques to investigate their morphologies, chemical environments, structures, microstructures, surface composition, optical features, bandgap, and magnetic performance. Specifically, scanning electron microscopy (SEM, TESCAN, MIRA) equipped with an electron-dispersive spectroscopy (EDS) detector was used to observe the morphologies and chemical environments, while X-ray diffraction (XRD, MiniFlex-600, Rigaku, Tokyo, Japan) and high-resolution transmission electron microscopy (HRTEM, JEM-2100F, JEOL) were used to analyze the structures and microstructures, respectively. X-ray photoelectronic spectroscopy (XPS, ESCALAB-250XI, Thermo Fisher, Waltham, MA, USA) was used to study the surface composition. A photoluminescence spectroscopy (PL, Cary Eclipse, Varian, Cheadle, UK) and UV–vis spectrometer (UV, PerkinElmer (Houston, TX, USA), Lambda 950) were used to analyze the optical features and bandgap, respectively. A vibrating sample magnetometer (VSM, Lake Shore (Westerville, OH, USA), 7404) was used to evaluate the magnetic performance.

### 3.6. Photocatalytic Measurement

The photocatalytic performance test was conducted under a xenon lamp source (PLS-SXE300) with a power of 300 W for the degradation of MO (10 mg/L) by Ag_2_O/Fe_3_O_4_. In the experiment, 100 mg of Ag_2_O/Fe_3_O_4_ was dispersed in 100 mL MO solution. After mixing, the mixture was stirred in the dark for 30 min to allow the catalyst to reach adsorption–desorption equilibrium with MO. The mixture containing the photocatalyst was then placed 10 cm away from the xenon lamp source and stirred at a speed of 200 r/min. During the light irradiation, 3 mL of the solution was taken out every 5 min and transferred to a centrifuge tube, and the catalyst powder was removed using a needle filter with a 0.22 μm pore size. A UV–visible spectrophotometer was used to measure the filtered MO concentration. The degradation rate of MO can be expressed as (C0-C)/C0, where C represents the MO concentration after xenon lamp irradiation, C0 represents the original concentration before irradiation, and the concentration of undegraded MO can be expressed as C/C0.

### 3.7. Photoelectrochemical Measurement

The experiment was conducted using an electrochemical analyzer (CHI660E, Shanghai) equipped with a standard three-electrode system. A 100 mL Na_2_SO_4_ solution (0.1 M) was used as the electrolyte, with a platinum (Pt) foil as the counter electrode, Ag/AgCl as the reference electrode, and the loading samples on FTO glass as working electrodes. Electrochemical impedance spectroscopy (EIS), a Mott–Schottky curve, and photocurrent response tests were performed.

## 4. Conclusions

In summary, a novel type of environmentally friendly magnetic nanocomposite, i.e., Ag_2_O/Fe_3_O_4_, has been synthesized and characterized as a high-performance visible-light-responsive photocatalyst. According to the XRD, SEM, TEM, XPS, UV–vis, PL, and electrochemical characterization, it has been confirmed that Ag_2_O and Fe_3_O_4_ are well compounded and exhibit a good synergistic effect. Loading Fe_3_O_4_ onto the surface of Ag_2_O not only constructs the type II heterojunction but also successfully couples the photocatalysis and Fenton reaction, enhancing its broad-spectrum response and efficiency. Under simulated sunlight irradiation, the Ag_2_O/Fe_3_O_4_ (10%) exhibited the fastest MO degradation rate, rapidly degrading 99.5% of 10 mg/L MO within 15 min, which was 2.4 times higher than that of pure Ag_2_O. Furthermore, after four cycles of testing, the sample still exhibited a fast degradation rate, indicating high stability. Magnetic testing emphasized the ease of material recovery using magnetic force, making the nanocomposite suitable for practical applications in water treatment and environmental remediation. Therefore, Ag_2_O/Fe_3_O_4_ exhibits magnetic properties, wide spectral response, and high oxidative degradation performance, and its preparation method provides a new approach for the development of future photocatalysts.

## Figures and Tables

**Figure 1 molecules-28-04155-f001:**
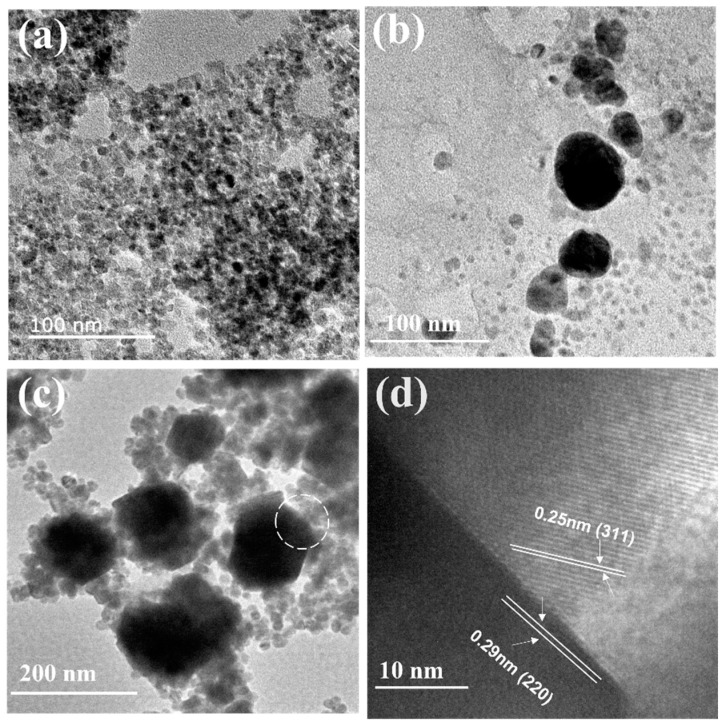
(**a**) TEM image of Fe_3_O_4_ nanoparticles; (**b**) TEM image of Ag_2_O nanoparticles; (**c**) TEM image of Ag_2_O/Fe_3_O_4_ (10%); (**d**) HRTEM image of Ag_2_O/Fe_3_O_4_ (10%).

**Figure 2 molecules-28-04155-f002:**
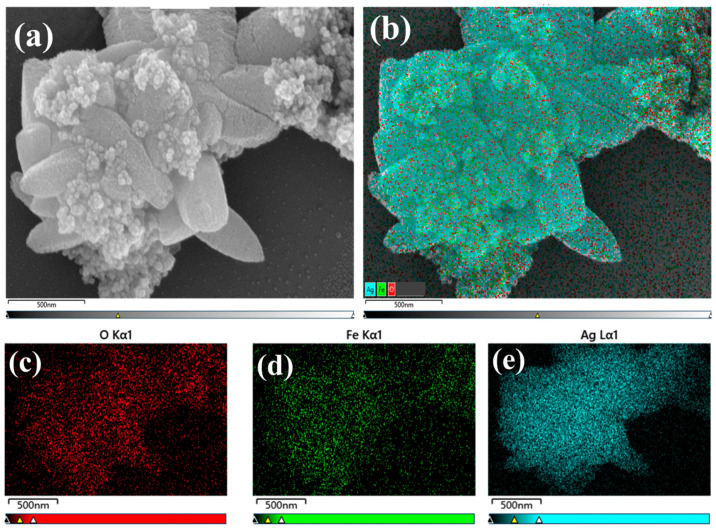
SEM images and element distributions of Ag_2_O/Fe_3_O_4_ (10%) nanoparticles, including (**a**) SEM image, (**b**) element distribution, and (**c**–**e**) O, Fe, and Ag element distributions.

**Figure 3 molecules-28-04155-f003:**
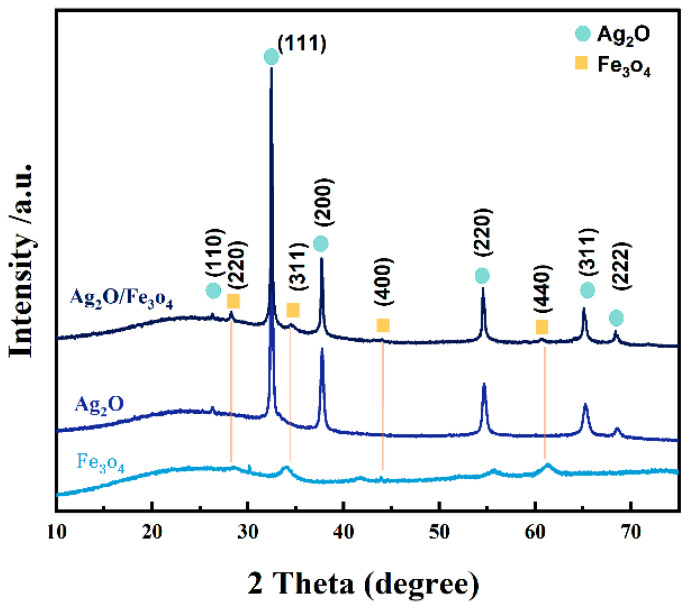
XRD characterization patterns of Ag_2_O/Fe_3_O_4_ (10%) nanoparticles, Ag_2_O nanoparticles, and Fe_3_O_4_ nanoparticles.

**Figure 4 molecules-28-04155-f004:**
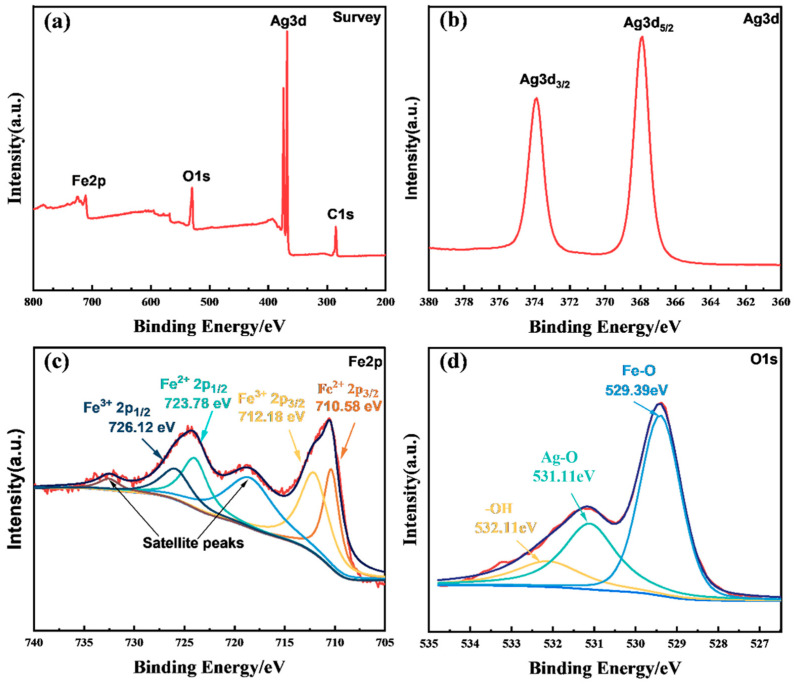
XPS spectra of Ag_2_O/Fe_3_O_4_ (10%), including (**a**) full-spectrum, (**b**) Ag3d high-resolution spectrum, (**c**) Fe2p high-resolution spectrum, and (**d**) O1s high-resolution spectrum.

**Figure 5 molecules-28-04155-f005:**
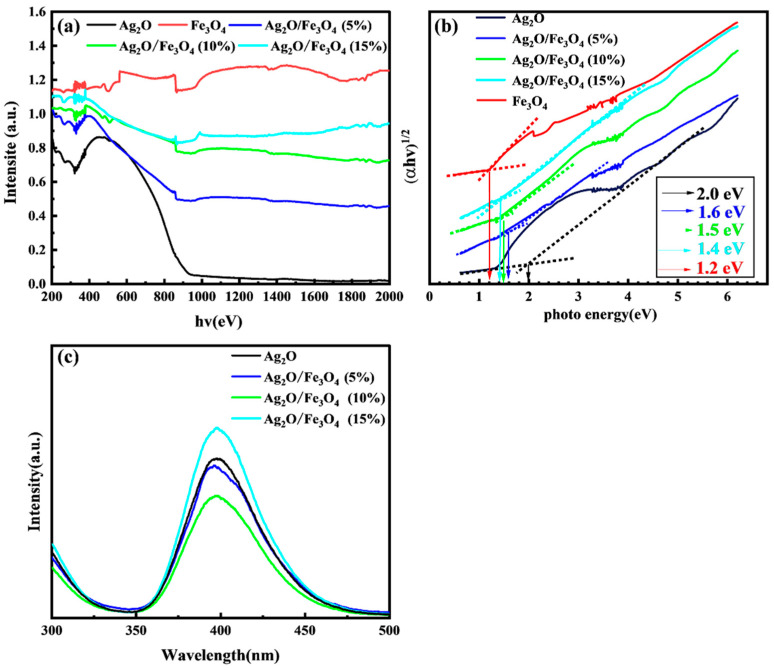
Ag_2_O, Fe_3_O_4_, Ag_2_O/Fe_3_O_4_ (5%), Ag_2_O/Fe_3_O_4_ (10%), and Ag_2_O/Fe_3_O_4_ (15%) of (**a**) UV–vis spectra and (**b**) Tauc plots. (**c**) PL spectra of Ag_2_O, Ag_2_O/Fe_3_O_4_ (5%), Ag_2_O/Fe_3_O_4_ (10%), and Ag_2_O/Fe_3_O_4_ (15%).

**Figure 6 molecules-28-04155-f006:**
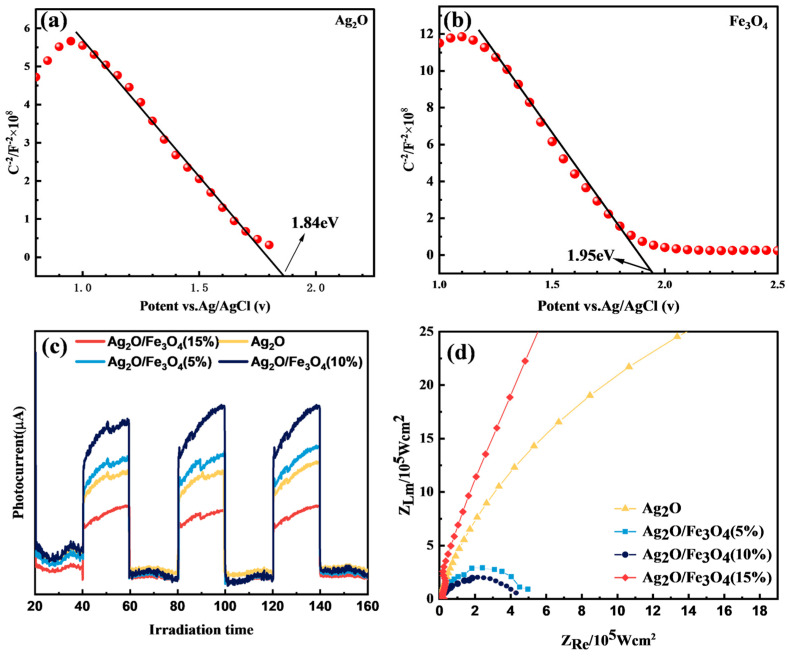
Mott–Schottky curves of (**a**) Ag_2_O and (**b**) Fe_3_O_4_; (**c**) sample photocurrent response profiles; (**d**) sample EIS test graph.

**Figure 7 molecules-28-04155-f007:**
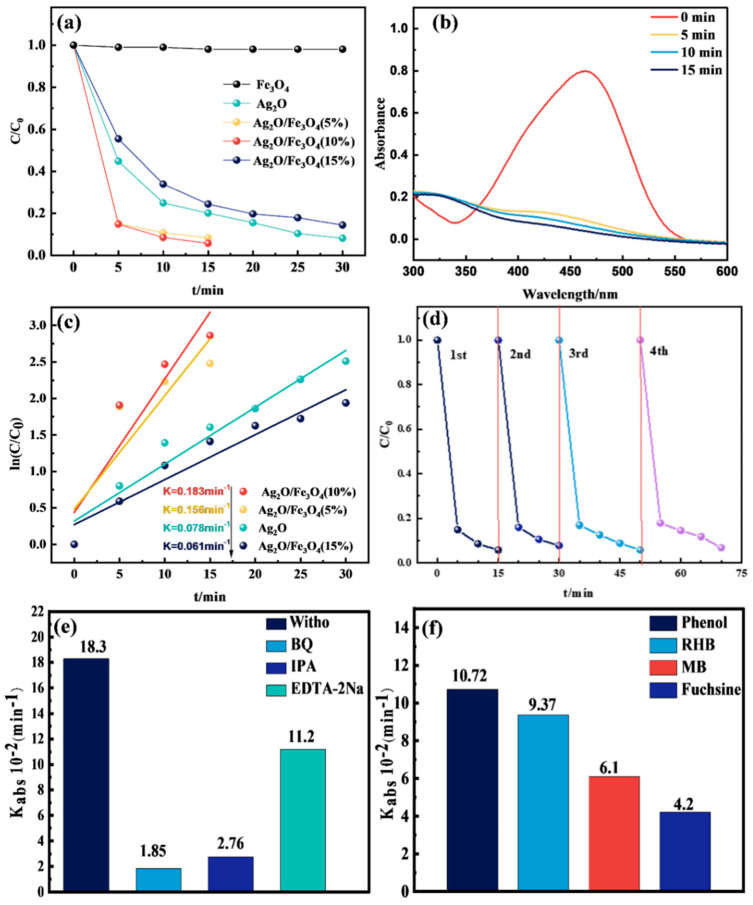
(**a**) Degradation rate of the MO solution by different photocatalyst samples; (**b**) UV–vis absorption spectra of the MO solution degraded by Ag_2_O/Fe_3_O_4_ (10%); (**c**) pseudo-first-order kinetic model diagram of the MO solution degradation; (**d**) cycle test of the Ag_2_O/Fe_3_O_4_ (10%) sample degradation; (**e**) kinetics rate of the MO solution degradation using Ag_2_O/Fe_3_O_4_ (10%) with different scavengers; (**f**) kinetics of different water pollutants degradation using Ag_2_O/Fe_3_O_4_ (10%).

**Figure 8 molecules-28-04155-f008:**
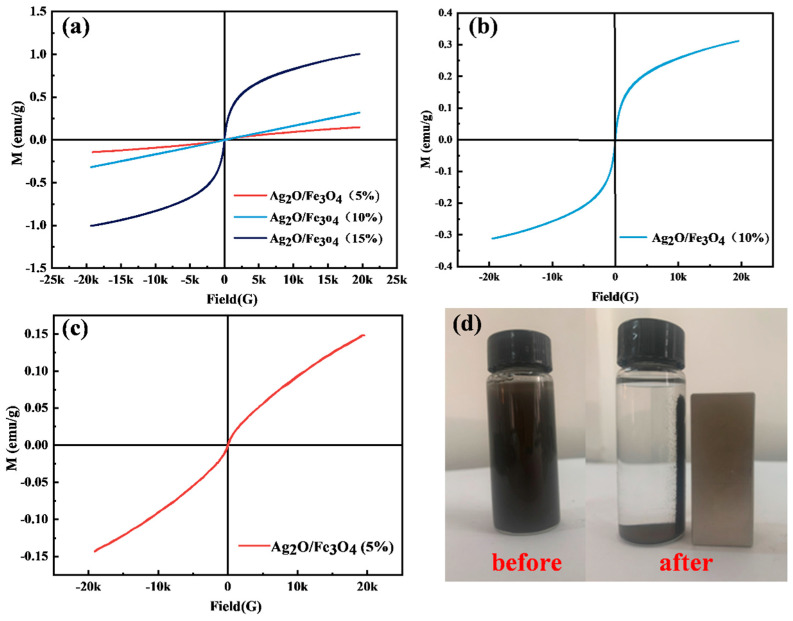
VSM curves of the (**a**) overall magnetic samples, (**b**) Ag_2_O/Fe_3_O_4_ (5%) composite, (**c**) Ag_2_O/Fe_3_O_4_ (10%) composite, and (**d**) test of Ag_2_O/Fe_3_O_4_ (10%) composite under Nd magnet adsorption.

**Figure 9 molecules-28-04155-f009:**
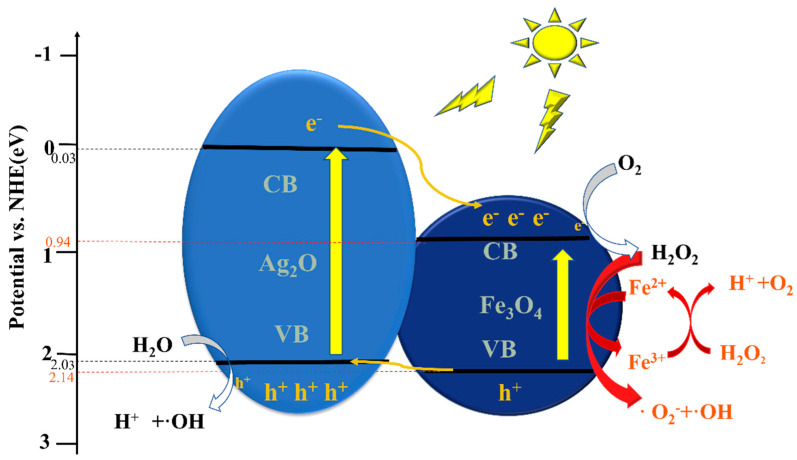
Schematic of the possible photocatalytic reaction mechanism of Ag_2_O/Fe_3_O_4_ under visible light irradiation.

**Table 1 molecules-28-04155-t001:** EDS data statistics of Ag_2_O/Fe_3_O_4_ (10%) nanoparticles.

Element	Line Type	wt%	wt% Sigma	at%
O	K series	21.33	0.46	63.01
Fe	K series	6.17	0.23	5.22
Ag	L series	72.49	0.47	31.76
Total		100.00		100.00

## Data Availability

The data can be made available upon reasonable request.

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
