# Peer review of "One-Step Synthesis of Ag2O/Fe3O4 Magnetic Photocatalyst for Efficient Organic Pollutant Removal via Wide-Spectral-Response Photocatalysis–Fenton Coupling"

_molecules, 2023, doi:10.3390/molecules28104155_

Round 1
Reviewer 1 Report
In the current work, the authors reported on the fabrication of a promising novel and environmentally friendly Ag2O/Fe3O4 (10 %) magnetic nanocomposite through coprecipitation. Moreover, photocatalytic performances of the nanocomposite were evaluated by degrading methyl orange. Collectively, the study is attractive and valuable for the scientific community, but several problems affect the overall quality of the manuscript, making it difficult to revise. The main issue is related to the lack of references shown in the text. Without direct links between the references section and the main text, it is hard to revise the study. Above all, please add specific references in the results and discussion section. Therefore, the authors should pay more attention to the format of the entire manuscript.
Based on this I recommend reconsidering it after major revisions.
Following a few comments are reported:
1. Abstract: Consider starting the abstract with a sentence about the benefits that the use of Ag2O/Fe3O4 can provide to the environment.
2. Figure 5: Please check out both sizes and dimensions. The figure exceeds the edges of the page. Anyway, the authors should carefully check captions, colors, sizes, and dimensions for all figures.
3. Equation 11: Check out the format.
4. Conclusions: Rather than summarizing the results section, the Conclusion section should conclude what this means for the overall process and its application in the real world. This section is similar to the abstract section. Please consider modifying it.
Author Response
First of all, I would like to express our sincere gratitude to the reviewers for their comments. These comments are all valuable and helpful for revising and improving our manuscript, as well as the important guiding significance to our researches. We have studied comments carefully and have made correction which we hope meet with approval. Revised portions are marked in red in the revised version. The corrections and responses to the reviewer's comments are listed below.
Responses to reviewers are in blue color.
Reviewer1:
In the current work, the authors reported on the fabrication of a promising novel and environmentally friendly Ag2O/Fe3O4 (10%) magnetic nanocomposite through coprecipitation. Moreover, photocatalytic performances of the nanocomposite were evaluated by degrading methyl orange. Collectively, the study is attractive and valuable for the scientific community, but several problems affect the overall quality of the manuscript, making it difficult to revise. The main issue is related to the lack of references shown in the text. Without direct links between the references section and the main text, it is hard to revise the study. Above all, please add specific references in the results and discussion section. Therefore, the authors should pay more attention to the format of the entire manuscript.
Based on this I recommend reconsidering it after major revisions.
Reply: Thank you for your suggestion. We have revised the manuscript accordingly and resubmitted it for your consideration.
Following a few comments are reported:
- Abstract: Consider starting the abstract with a sentence about the benefits that the use of Ag2O/Fe3O4 can provide to the environment.
Reply: Thank you for your suggestion. We have revised the abstract to emphasize the environmental benefits of using Ag2O/Fe3O4 as an efficient photocatalyst. We believe the updated abstract now better reflects the importance of this nanocomposite for addressing environmental challenges.
- Figure 5: Please check out both sizes and dimensions. The figure exceeds the edges of the page. Anyway, the authors should carefully check captions, colors, sizes, and dimensions for all figures.
Reply: We appreciate your feedback on Figure 5. We have corrected its size and dimensions, and reviewed all other figures to ensure proper presentation.
- Equation 11: Check out the format.
Reply: Thank you for pointing out the issue with Equation 11. We have carefully checked and corrected its format to ensure consistency and clarity.
- Conclusions: Rather than summarizing the results section, the Conclusion section should conclude what this means for the overall process and its application in the real world. This section is similar to the abstract section. Please consider modifying it.
Reply: Thank you for your suggestion. We have carefully reviewed your suggestion and have made the necessary modifications to the Conclusion section of the paper. The section now concludes what the results mean for the overall process and its potential real-world applications, as opposed to just summarizing the results. We appreciate your insightful feedback, as it has helped to improve the quality and clarity of the paper.
Reviewer 2 Report
the paper report One-Step Synthesis of Ag2O/Fe3O4 Magnetic Photocatalyst for Efficient Organic Pollutant Removal via Wide Spectral Response Photocatalysis-Fenton Coupling work is good following changes need to be- done.
1. In the abstract background and important finding must be given.
2. There are other method by which organic pollutant can be removed using photocatalyst author may go through https://doi.org/10.1155/2022/3970287
3. How much is the phocatalytic activity had author compmared with any standard compound.
4. Check for spelling and english errors
5. References should be according to journal format.
Check for spelling and english errors
Author Response
The paper report One-Step Synthesis of Ag2O/Fe3O4 Magnetic Photocatalyst for Efficient Organic Pollutant Removal via Wide Spectral Response Photocatalysis-Fenton Coupling work is good following changes need to be- done.
- In the abstract background and important finding must be given.
Reply: Thank you for your feedback. We have carefully reviewed the abstract and have made the necessary changes to include a brief background and important finding. We appreciate your suggestion and thank you for helping us improve the quality of our work.
- There are other method by which organic pollutant can be removed using photocatalyst author may go through https://doi.org/10.1155/2022/3970287
Reply: Thankes for your suggestion and the provided article. We have carefully read the article and considered the possibility of other mechanisms for the removal of organic dyes using photocatalysts, cited you recommended articles (as Ref.39) In our study, as described in Section 3.6 "Photocatalytic measurement," we ensured that the all photocatalysts reached adsorption-desorption equilibrium with MO in the dark after 30 minutes. This approach minimizes the impact of adsorption and other non-photocatalytic reactions on the degradation rate.
Additionally, MO is known to be a stable organic dye that does not undergo significant degradation under light exposure in the absence of a photocatalyst. Therefore, our experimental design effectively excludes the influence of factors other than photocatalysis and Fenton reactions on the degradation of organic dyes in our study.
We hope this response addresses your concern.
- How much is the phocatalytic activity had author compmared with any standard compound.
Reply: In response to your question on photocatalytic activity comparison with a standard compound, we have compared the photocatalytic activity of our Ag2O / Fe3O4 nanocomposite with both pure Ag2O and Fe3O4. As shown in Figure 7a, when pure Fe3O4 was placed in the MO solution and irradiated with visible light, no MO degradation was observed, indicating that Fe3O4 alone does not have the ability to degrade the MO solution under visible light. In contrast, pure Ag2O displayed significant degradation of MO under visible light, with approximately 80% degradation in 15 min and 99.1% in 30 min.Figure 7c further demonstrates that the degradation rate of Ag2O/Fe3O4 (10%) is the fastest at 0.183 min-1, which is significantly higher than that of Fe3O4, 2.3 times higher than that of pure Ag2O (0.078 min-1), 3 times higher than that of Ag2O / Fe3O4 (15%) (0.061 min-1), and 1.17 times higher than that of Ag2O/Fe3O4 (5%) (0.156 min-1). This comparison highlights the enhanced photocatalytic activity of our Ag2O/Fe3O4 nanocomposite compared to both Fe3O4 and Ag2O.
We hope this response addresses your concern.
- Check for spelling and english errors
Reply: I have checked the manuscript for spelling and English errors and have made the necessary corrections. Thank you for bringing this to my attention.
- References should be according to journal format.
Reply: Thank you for your suggestion regarding the references. We have reviewed and updated the reference section according to the journal format. We appreciate your careful review and helpful feedback.
Reviewer 3 Report
Review Report
The work "One-Step Synthesis of Ag2O/Fe3O4 Magnetic Photocatalyst for Efficient Organic Pollutant Removal via Wide Spectral Response Photocatalysis-Fenton Coupling by Minor Alteration of N-Alkylation" by Chuanfu Shan et al. presents a novel, stable, and environmentally friendly Ag2O/Fe3O4 magnetic nanocomposite synthesized via simple coprecipitation. The method proposed by authors offers a new approach in constructing high-performance photocatalytic systems.
The paper adds valuable data and methodology in the field of synthesis of Ag2O/Fe3O4 magnetic photocatalyst.
The paper reads well and the study is clear and complete. The work is timely for the for the Molecules community. I recommend to publish it in a present form.
Author Response
The work "One-Step Synthesis of Ag2O/Fe3O4 Magnetic Photocatalyst for Efficient Organic Pollutant Removal via Wide Spectral Response Photocatalysis-Fenton Coupling by Minor Alteration of N-Alkylation" by Chuanfu Shan et al. presents a novel, stable, and environmentally friendly Ag2O/Fe3O4 magnetic nanocomposite synthesized via simple coprecipitation. The method proposed by authors offers a new approach in constructing high-performance photocatalytic systems.The paper adds valuable data and methodology in the field of synthesis of Ag2O/Fe3O4 magnetic photocatalyst.The paper reads well and the study is clear and complete. The work is timely for the for the Molecules community. I recommend to publish it in a present form.
Reply: Thank you for taking the time to review our manuscript. We are grateful for your positive comments on our work and for recognizing the novelty and importance of our approach in constructing high-performance photocatalytic systems.
Reviewer 4 Report
Although this topic is very interesting, however, in this form, this manuscript is not suitable for publication and requires very serious revision.
1. In the attached pdf of this manuscript, the introduction does not contain links to the cited articles but shows an error. This means that the authors did not look at the final version of the article and this is not good.
2. the lack of links to the cited articles does not allow us to fully assess the quality of the introduction.
3. However, it is clear that the information on Fe3O42 given here is not quite detailed. Additional information about its functional properties, which depend on its preparation, size and morphology, will be very important and useful. 1. This will make the article more interesting and attractive to a wide range of readers. See for example few recent MDPI published research and references therein:
Serga, V.; Burve, R.; Maiorov, M.; et al. Impact of Gadolinium on the Structure and Magnetic Properties of Nanocrystalline Powders of Iron Oxides Produced by the Extraction-Pyrolytic Method. Materials 2020, 13, 4147. https://doi.org/10.3390/ma13184147
Nordin, A.H.; Ahmad, Z.; et al. The State of the Art of Natural Polymer Functionalized Fe3O4 Magnetic Nanoparticle Composites for Drug Delivery Applications: A Review. Gels 2023, 9, 121. https://doi.org/10.3390/gels9020121
4. Fig.1. Were there any effects associated with aging during those measurements or simply with time after preparation for samples containing Ag2O?
5. Table 1. Such EDS data statistics need error bars!
6. Paragraph 3.5. It is necessary to provide literary data for bulk Fe3O4 and Ag2O compounds.
7. Fig.5b. Can one still justify drawing a straight line to determine Eg. The definition of the band gap Eg here is quite ambiguous. See the latest article by the Editors of Optical Material (Elsevier): M.G. Brik, A.M. Srivastava et al, A few common misconceptions in the interpretation of experimental spectroscopic data. Optical Materials 127 (2022) 112276 https://doi.org/10.1016/j.optmat.2022.112276.
8. In the case of analysis of the optical properties, what is the actual concentration of point defects (oxygen vacancies), and how they were taken into account in the analysis of band gap parameters?
9. Fig.5C. The text does not provide an explanation of the mechanism of luminescence.
MAJOR REVISION.
Author Response
Although this topic is very interesting, however, in this form, this manuscript is not suitable for publication and requires very serious revision.
- In the attached pdf of this manuscript, the introduction does not contain links to the cited articles but shows an error. This means that the authors did not look at the final version of the article and this is not good.
Reply: Thank you for bringing this to our attention. We apologize for the error in the PDF version of our manuscript. We have checked the manuscript and made sure that all links are properly inserted in the final version of the article.
- the lack of links to the cited articles does not allow us to fully assess the quality of the introduction.
Reply: Thank you for bringing this to our attention. We apologize for the oversight in not including the links to the cited articles. We have reviewed and revised the manuscript to include the appropriate links to the references as per the journal's guidelines. We appreciate your valuable feedback and will ensure to thoroughly review and check the final version before submission.
- However, it is clear that the information on Fe3O4 given here is not quite detailed. Additional information about its functional properties, which depend on its preparation, size and morphology, will be very important and useful. 1. This will make the article more interesting and attractive to a wide range of readers. See for example few recent MDPI published research and references therein:
Serga, V.; Burve, R.; Maiorov, M.; et al. Impact of Gadolinium on the Structure and Magnetic Properties of Nanocrystalline Powders of Iron Oxides Produced by the Extraction-Pyrolytic Method. Materials 2020, 13, 4147. https://doi.org/10.3390/ma13184147
Nordin, A.H.; Ahmad, Z.; et al. The State of the Art of Natural Polymer Functionalized Fe3O4 Magnetic Nanoparticle Composites for Drug Delivery Applications: A Review. Gels 2023, 9, 121. https://doi.org/10.3390/gels9020121
Reply: Thankes for your suggestion and the provided article ,we would like to provide additional information regarding the preparation, size, morphology, and functional properties of Fe3O4 in our study.
In Section 3.2 "Preparation of Fe3O4" we described the process of preparing Fe3O4 nanoparticles. In Section 2.1 "TEM analysis," we reported that the average particle size of Fe3O4nanoparticles was 15 ± 5 nm with a typical spherical morphology. In Section 2.2 "SEM and EDS analysis," we discussed the compact structure of binary nanoparticles with good structural stability, as well as the increased number of reaction sites due to the loading of Fe3O4. In Section 2.9 "Photocatalytic reaction mechanism analysis," we addressed the Fenton reaction involving Fe3O4 and its role in the photocatalytic process.Moreover 2.8 "Magnetic properties analysis," where we demonstrated that Ag2O/ Fe3O4 (5%), Ag2O/ Fe3O4 (10%), and Ag2O/ Fe3O4 (15%) all have superparamagnetic properties due to the incorporation of Fe3O4. This feature further enhances the practical applicability of our photocatalyst.
We have carefully reviewed and cited the you recommended articles by Serga et al. (as the Ref. 22) and Nordin et al. (as the Ref. 17) to further enrich our understanding of the functional properties of Fe3O4 and its dependence on preparation, size, and morphology.
Thank you for your valuable input, which has greatly contributed to the improvement of our work.
- 4. Were there any effects associated with aging during those measurements or simply with time after preparation for samples containing Ag2O?
Reply: Thank you for your insightful question. In our TEM measurements presented in Figure 1, we did not observe any changes in the nanostructure or morphology of the Ag2O-containing samples due to variations in the time of preparation or testing. Furthermore, Ag2O and Fe3O4, as metal oxides, possess relatively stable physical properties and are not prone to aging or degradation. We appreciate your concern and hope this response addresses your query.
- Table 1. Such EDS data statistics need error bars!
Reply: In response to your concern, Table 1 does indeed include error bars. The "Wt % Sigma" column in the table represents the error bars for each element's weight percentage (wt%). By including these uncertainties associated with the measurements, the table provides the necessary information to assess the reliability of the data.
- Paragraph 3.5. It is necessary to provide literary data for bulk Fe3O4 and Ag2O compounds.
Reply: Thank you for your question regarding the literary data for bulk Fe3O4 and Ag2O compounds. In response, we have updated our manuscript, specifically in Paragraph 2.5, to include relevant references. We cited references [13, 14, 15] for Ag2O and [23, 31, 32, 33] for Fe3O4 in the revised paragraph. We believe that these references adequately address your concern and provide valuable insights into the properties of these compounds. Additionally, we have changed the original "Section 3" to " Section 2" at the editor's request
- Fig.5b. Can one still justify drawing a straight line to determine Eg. The definition of the band gap Eg here is quite ambiguous. See the latest article by the Editors of Optical Material (Elsevier): M.G. Brik, A.M. Srivastava et al, A few common misconceptions in the interpretation of experimental spectroscopic data. Optical Materials 127 (2022) 112276 https://doi.org/10.1016/j.optmat.2022.112276.
Reply: Thank you for pointing out the issue. We have carefully read the recommended literature and the referenced articles within it. This has helped us understand the differences between the two approaches, direct extrapolation (DE) and proper extrapolation (PE), for evaluating the bandgap.
In light of this new understanding, we acknowledge the limitations of using the DE method for determining Eg. Consequently, we have revised our manuscript to adopt the PE method for determining the bandgap and updated the relevant sections accordingly. We believe that this change addresses your concern and ensures a more accurate interpretation of our experimental data.
Figure 5. Ag2O, Fe3O4, Ag2O/Fe3O4 (5%), Ag2O/Fe3O4 (10%), and Ag2O/Fe3O4 (15%) of (a) UV–vis spectra and (b) Tauc plots.
- In the case of analysis of the optical properties, what is the actual concentration of point defects (oxygen vacancies), and how they were taken into account in the analysis of band gap parameters?
Reply: Thank you for your question .Oxygen vacancies refer to the positions of missing oxygen atoms in the lattice, leading to unsaturated metal bonds in the lattice. These unsaturated metal bonds can potentially adsorb water molecules upon contact with water or water vapor, resulting in the formation of -OH groups.
In our XPS measurements presented in Figure 4d shows the O 1s fine spectrum of Ag2O/Fe3O4 (10%). By calculating the overall proportion of the -OH group-related peak area in the O1s spectrum, we can estimate that the concentration of oxygen vacancies is approximately 13.4%. For Ag2O, oxygen vacancies may introduce defect states, affecting the bandgap. However, in practice, the effect of oxygen vacancies on the relatively simple lattice structure of Ag2O may be minimal. Therefore, when evaluating their bandgaps, the presence of oxygen vacancies can be neglected. For Fe3O4, due to its complex lattice structure and magnetism, oxygen vacancies may have a more significant impact on its bandgap. Oxygen vacancies may cause the bandgap to increase or decrease, depending on the position of the defect states within the energy bands. However, due to the experimental conditions limitations, it is not possible to determine the position of the oxygen vacancies in the energy bands, and the characterized oxygen vacancy concentration in Ag2O/Fe3O4 (10%) is relatively small. Therefore, in the analysis of bandgap parameters, the influence of oxygen vacancies on the bandgap is not discussed.
We hope this response addresses your concern.
- Fig.5C. The text does not provide an explanation of the mechanism of luminescence.
Reply: Thank you for bringing this to our attention. We have added the requested explanation of the PL mechanism related to the samples presented in Fig. 5c and explored the relationship between PL intensity and the separation of photogenerated charge carriers in the manuscript. The text mentions that when Ag2O and Fe3O4 are exposed to light, the valence band electrons absorb photon energy and transition to the conduction band, forming photogenerated electron-hole pairs. The PL emission occurs when the conduction band electrons recombine with valence band holes, and the lower PL intensity reflects a reduction in recombination probability due to the increased separation of the photogenerated charge carriers. Therefore, we believe that the text has provided an adequate explanation of the PL mechanism and its relation to the separation of photogenerated charge carriers.
Thank you for your valuable input, which has greatly contributed to the improvement of our work.
Round 2
Reviewer 1 Report
The authors carefully followed the reviewer's recommendations, which means that the manuscript can be considered for acceptance in the present form.
Reviewer 4 Report
After successful revision, this manuscript can be recommended for publication.